# OpenReview forum: "Detecting and Evaluating Medical Hallucinations in Large Vision Language Models"
_ICLR.cc/2025/Conference — Submitted to ICLR 2025_

### Official Review · Reviewer_Sam2 · 2024-10-30

**Soundness:** 2
**Presentation:** 3
**Contribution:** 2
**Rating:** 5
**Confidence:** 3

**Summary:**

The paper proposes an LLM/LVLM evaluation benchmark, especially for the hallucination in the medical domain tasks. It also proposes a metric to evaluate the severity of hallucination in LVLMs, and a fine-tuned evaluator model is released to perform the universal evaluation for given set information based on (input image I, the original prompt P, the LVLM answer A and the ground truth GT).

**Strengths:**

The paper proposes a novel benchmark for evaluation the LLMs specially in the Medical domain, and provides a comprehensive framework from dataset, metric, and the fine-tuned evaluation model, which is a completed work through the evaluation pipeline.
●	The paper is writing in a smooth way, easy to follow and simple to understand.
●	The work is of good significance, should be a meaningful angle to advance the LLM/LVLMs applications in real-world healthcare domain.

**Weaknesses:**

●	The paper did not emphasize the special challenges in Healthcare domain, after the reading, except for the first part of the dataset that involves the `Medical` multi-modality data such as CT, the evaluation process is on the common hallucination challenges from any Vision-language model, it raises the concerns that whether the proposed model stands out from common LVM hallucination evaluation, is other existing work able to solve the same question by simply adapting to the medical data? What is the advantage of the proposed fine-tuned model over other baseline methods?
●	The hierarchical definition of the MediHALL score seems a bit intuitive, which is a simple evenly scaled value on 5 categories.
●	The paper lacks discussion on the relevant work that conduct similar hallucination detection in LLMs at Healthcare domain.
●	It is suggested that the paper could make further clarification on certain questions as in the Question section, if most critical concerns can be addressed, it is plausible to raise the score for the paper’s quality evaluation.

**Questions:**

1. The authors explicitly defined 5 types of hallucination levels, among which: Catastrophic Hallucination, Critical Hallucinations, Minor Hallucination are severity levels, but the attribute, prompt-induced ones is indeed the `cause` of the hallucinations. There raise doubts of the rationale of such empirical classifications including attribute, prompt-induced to severity levels.

2. As in line 215, the conventional Q_{conv} questions are generated by the GPT3.5, how is the quality guaranteed? How is the initial conv question obtained?

3. LLaVA-Med was used to infer answers, what is the size of the whole dataset? Even though the authors claim that the data has been examined, how is the quality guaranteed?

4. Also, for the dataset part that is relevant to the IRG scenario, the authors use a sampling method to draw 1,800 images and their corresponding medical reports from existing datasets: MIMIC-test and OpenI datasets. This weakens the contribution towards the data as proposed in line 065.

5. In line 255, “This fine-grained metric xxx”, starts abruptly, since the previous paragraph discusses the existing metrics drawbacks. Then it directly refers to “This metric”, which is not transitioning naturally.

6. The MediHALL score is relatively simple, which is built upon the pre-categorized types of hallucination levels and assigns different values in a hierarchical way.  Based on this, the latter human-annotation and fine-tuning are conducted, are they able to maintain the objectiveness?

7. In line 318, which directs the training data details to Figure, it explicitly shows the categories, and the types of training data covered for MEDIHALLDETECTOR, but what about the amount of the data? And how is the instruction pair data derived?

8. While the paper is positioned as the first benchmark for medical LM hallucination evaluation, there is relevant work worth referencing:

[1] MEDHALU: Hallucinations in Responses to Healthcare Queries by Large Language Models
[2] MEDIC: TOWARDS A COMPREHENSIVE FRAMEWORK FOR EVALUATING LLMS IN CLINICAL APPLICATIONS
[3] Faithfulness Hallucination Detection in Healthcare AI

---

### Official Review · Reviewer_h5iV · 2024-11-02

**Soundness:** 3
**Presentation:** 3
**Contribution:** 2
**Rating:** 3
**Confidence:** 5

**Summary:**

The paper introduces a new benchmark, Med-HallMark, and proposes evaluation tools specifically tailored to medical contexts. Med-HallMark provides a multi-dimensional framework to identify and assess hallucinations generated by LVLMs. The benchmark includes multi-tasking hallucination support, diverse data for hallucination detection, and a hierarchical categorization of hallucinations based on clinical impact. Additionally, the paper presents MediHall Score, a metric designed to evaluate hallucinations in medical text output, and MediHallDetector, a specialized LVLM aimed at improved hallucination detection through multi-task training.

**Strengths:**

- The introduction of Med-HallMark and MediHall Score fills a significant gap by addressing hallucination detection in medical LVLMs.
- The proposed categorization of hallucinations is innovative and highly relevant, allowing for analysis of potential model impacts on medical decision-making.
- The authors conduct extensive experiments comparing popular LVLM models on Med-HallMark.

**Weaknesses:**

- Is this very different from the accuracy? After reading the paper, I just feel that this paper just makes a more fine-grained classification of errors, and the classification standard needs to be discussed and re-designed. For example, in the report generation task, the model outputs a nonsense sentence for the chest X-ray, "This is a chest X-ray of a person", which is obviously correct, but it is not what we want. How should this be judged?
So this makes me wonder whether such a benchmark is necessary.
- Some recent related work [1,2,3,4,5,6] is missing.
- The overall data scale is relatively small, and the medical image modalities involved are limited to radiology.
- How to ensure the accuracy of annotation?
- Since doctors are hired to do the annotation, have the possible ethical risks been resolved? For example, IRB approval, etc.



[1] Gu Z, Yin C, Liu F, et al. MedVH: Towards Systematic Evaluation of Hallucination for Large Vision Language Models in the Medical Context[J]. arXiv preprint arXiv:2407.02730, 2024.

[2] Yan Q, He X, Wang X E. Med-HVL: Automatic Medical Domain Hallucination Evaluation for Large Vision-Language Models[C]//AAAI 2024 Spring Symposium on Clinical Foundation Models. 2024.

[3] Jiang Y, Chen J, Yang D, et al. MedThink: Inducing Medical Large-scale Visual Language Models to Hallucinate Less by Thinking More[J]. arXiv preprint arXiv:2406.11451, 2024.

[4] Xia P, Chen Z, Tian J, et al. CARES: A Comprehensive Benchmark of Trustworthiness in Medical Vision Language Models[J]. arXiv preprint arXiv:2406.06007, 2024.

[5] Nan Y, Zhou H, Xing X, et al. Beyond the Hype: A dispassionate look at vision-language models in medical scenario[J]. arXiv preprint arXiv:2408.08704, 2024.

[6] Yan Q, He X, Yue X, et al. Worse than Random? An Embarrassingly Simple Probing Evaluation of Large Multimodal Models in Medical VQA[J]. arXiv preprint arXiv:2405.20421, 2024.

**Questions:**

- The format of the reference is weird. Please check it.
- The figure 2 (d) is vague.

**Details Of Ethics Concerns:**

Since doctors are hired to do the annotation, have the possible ethical risks been resolved? For example, IRB approval, etc.

---

### Official Review · Reviewer_69eN · 2024-11-02

**Soundness:** 3
**Presentation:** 3
**Contribution:** 2
**Rating:** 6
**Confidence:** 3

**Summary:**

The paper proposes Med-HallMark, a benchmark to assess hallucination detection in large vision-language models (LVLMs) applied to medicine. This paper suggests MediHallDetector as the first hallucination detection model able to evaluate medical image report generation and visual question answering, and outperforming baselines.
The authors also proposes an evaluation metric, MediHall Score to gauge the level of hallucination in LVLMs’ answers. Both qualitative and quantitative results are provided to show the superiority of MediHallDetector over baseline models.

**Strengths:**

•	Novelty of Benchmark Proposal: the medical benchmark is novel and relevant in assessing LVLMs hallucinations

•	Robustness: Test time evaluations with several comparisons on annotated sets strengthen the validity of the approach.

•	Evaluation: Baselines are carefully chosen and evaluated.

•	Clinical Impact: This study includes expert annotators which makes the dataset reliable and applicable.

**Weaknesses:**

•	Reproducibility: The authors highlight the benefit of hierarchical categorization but the annotation process is not fully explained for generalization purposes.

•	Evaluation: Lack of discussion for the reasons behind the difference in performance of the models employed.

•	The proposed benchmark includes two modalities X-ray and CT and questions remain on whether it could be applied and work for other imaging modalities.

**Questions:**

•	Did you encounter failures ? Could you elaborate on those cases where MediHallDetector fails and what are the causes.

•	Do you think MediHallDetector could work on other multimodal medical data ? What are the potential challenges and difficulties in extending MediHallDetector to out-of-domain data?

•	Given the rapid evolution of LVLMs, does MediHallDetector need to be updated ?

---

### Official Review · Reviewer_CLg1 · 2024-11-04

**Soundness:** 2
**Presentation:** 3
**Contribution:** 2
**Rating:** 3
**Confidence:** 4

**Summary:**

This paper introduces a hallucination detection and evaluation benchmark in the medical multimodal domain. Recognizing that large vision-language models often inherit hallucination tendencies from foundational large-language models, the authors propose a framework for more accurate and detailed hallucination assessment in high-stakes medical applications.
The paper makes three primary contributions: (i) Med-HallMark, the benchmark dedicated to medical hallucination detection (ii) MediHallDetector, a hallucination detection model tailored for the medical domain; and (iii) MediHall Score, an evaluative metric designed for hallucinations of different severity and types.
Through extensive experiments, the paper claims that MediHall Score provides more nuanced insights than traditional metrics and that MediHallDetector enhances hallucination detection performance in medical LVLMs.

**Strengths:**

The paper is well-motivated in that it addresses a critical and underexplored area in medical LVLMs: detecting and evaluating hallucinations, which is crucial for ensuring safety in clinical applications. The work frames hallucination detection specifically for the medical domain. The MediHall Score introduces a nuanced metric that prioritizes clinical impact by differentiating hallucinations based on their severity and type. The ablation study of different SFT methods for training the detector also adds robustness to the study.

**Weaknesses:**

The paper raises several questions about primarily the soundness of evaluation and the reliability of the proposed metric (detailed in the **Questions** section). For example, the paper limits its IRG evaluation to only one domain-specific model, XrayGPT, which is problematic given that IRG tasks are where hallucination metrics are particularly valuable. Additionally, results for XrayGPT in IRG appear inconsistent with those of Minigpt-4, which was shown to underperform in traditional metrics but achieved the highest MediHall Score among all evaluated models, calling into question the reliability of the MediHall Score. The paper could have strengthened its findings by evaluating other state-of-the-art LVLMs, such as open-source, domain-specific models like BiomedGPT or Med-Flamingo, or proprietary MLLMs like GPT-4 and Gemini Pro, to validate the metric’s robustness. Finally, the experimental design lacks a thorough investigation into the role of image inputs in tuning MediHallDetector. The study could benefit from ablation studies that assess performance with and without image inputs, especially given that the image encoder and connector taken from LLaVA are frozen during training and could have introduced extra errors due to its lack of visual understanding ability. These design choices suggest that the current benchmark version may be limited in scope and could benefit from further refinement and validation.

**Questions:**

**Construction of the benchmark:**
1. Could you elaborate on the motivation behind designing confidence-weakening and counterfactual questions beyond the "conventional" questions in established test sets like SLAKE and RAD-VQA? Regarding counterfactual question generation, considering the GPT model is provided with limited information (text-only, single question with its ground-truth answer), how is the quality of GPT-generated counterfactual questions and answers ensured?

2. Robust evaluation of hallucinations in open-ended IRG tasks is inherently challenging, particularly due to synonymous terms (in frameworks like ICD-10, multiple expressions might refer to the same condition). Since the ground truth for open-ended IRG scenarios in this paper derives from medical reports in MIMIC and OpenI test sets, how does the evaluation framework account for potential synonyms in model responses given the limited set of 1800 images and their reports seen by MediHallDetector?

3. In using LLaVA-Med to generate GT responses for Med-VQA, why not utilize the GT already provided in the dataset, given that the questions originate from an established source? For the IRG scenario, are model-generated responses also exclusively from LLaVA-Med? If so, might this narrow distribution affect the generalizability of MediHallDetector, as it is fine-tuned specifically on LLaVA-Med responses?

**Experimental Design:**
1. How much does the image come into play in tuning the MedihallDetector model? It could be necessary to ablate w./w.o. image input, especially when you freeze the image encoder and connector during training.

**Results**
1. For medical VQA tasks (Table 1), why introduce a MediHall Score when accuracy already exists as a metric for Med-VQA? Notably, XrayGPT, with an accuracy of only 0.02, has a MediHall Score of 0.36. Could this indicate potential inflation in the MediHall Score?

2. For medical IRG tasks (Table 2), the results for XrayGPT as measured by MediHallDetector seem counter-intuitive compared to Minigpt-4. In the original XrayGPT study [1], Minigpt-4 was a baseline against which improvements were demonstrated using ROUGE (Table 1 in [1]). Here, XrayGPT does outperform Minigpt-4 in all conventional metrics, but in the MediHall Score, with Minigpt-4 obtaining the highest MediHall Score among evaluated models. Could this raise concerns regarding the reliability of the MediHall Score metric?

3. Could you clarify why models like BLIP2, LLaVA-Med, and RadFM do not receive MediHall Scores for medical IRG tasks? Open-ended IRG tasks are particularly significant since accuracy is already used as a metric for VQA. Presently, only XrayGPT is evaluated in IRG tasks using the proposed model and metric; incorporating additional state-of-the-art models would enhance the proposed metric's robustness. For instance, open-source domain-specific models such as BiomedGPT [2] and Med-Flamingo [3], which perform well on established tasks, or widely used proprietary MLLMs like GPT-4, Claude 3.5, and Gemini Pro could be valuable benchmarks.

**Typo**
1. In Figure 2(a), “pairs” is misspelled.
2. Could you clarify whether XrayGPT is used in constructing the benchmark? It appears in Figure 1(b) but is not referenced in section 3.4.

[1] https://aclanthology.org/2024.bionlp-1.35.pdf
[2] https://arxiv.org/abs/2305.17100
[3] https://arxiv.org/abs/2307.15189

---

### Meta-Review · Area_Chair_3MJE · 2024-12-19

**Metareview:**

This paper introduces Med-HallMark, a benchmark for detecting and evaluating hallucinations in medical multimodal tasks, along with MediHallDetector, a fine-tuned hallucination detection model, and MediHall Score, a nuanced evaluation metric, collectively advancing the reliability and assessment of large vision-language models in high-stakes medical applications.

Reviewers agreed that the strengths of this paper include its introduction of Med-HallMark and MediHall Score, addressing a critical gap in hallucination detection for medical LVLMs. Meanwhile, Expert-annotated data, nuanced categorization and robust evaluations also contribute to advancing LVLM applications in healthcare.   On the other hand, several common concerns were raised by reviewers, focusing on evaluation soundness, dataset limitations, and reproducibility. Reviewers CLg1, 69eN, and h5iV noted the limited dataset size and restriction to X-ray and CT modalities, which hinder the benchmark's generalizability to other medical domains. The reliability of the MediHall Score was questioned by Reviewers CLg1, 69eN, and Sam2, citing inconsistent results and an overly intuitive hierarchical definition lacking justification. Reviewers h5iV and Sam2 also highlighted the lack of discussion on related work in hallucination detection for healthcare-specific LLMs and insufficient comparisons with existing methods. Additionally, Reviewers 69eN and h5iV criticized the annotation process for being inadequately detailed, raising concerns about reproducibility and ethical considerations such as IRB approval.

No rebuttal was submitted.

Three reviewers leaned towards rejection while one reviewer suggested to be marginally above the acceptance threshold. After carefully considering the reviewers' comments and the absence of a rebuttal from the authors, I have decided to reject the paper. While the paper introduces a novel benchmark and metric for hallucination detection in medical LVLMs, significant concerns remain unresolved. These include the limited dataset size and scope, inconsistent evaluation results, questions about the reliability and design of the MediHall Score, and a lack of detail in the annotation process, raising reproducibility and ethical concerns. Without a rebuttal to address these critical issues, the paper does not meet the standard required for acceptance.

**Additional Comments On Reviewer Discussion:**

Reviewers CLg1, 69eN, and h5iV noted the limited dataset size and restriction to X-ray and CT modalities, which hinder the benchmark's generalizability to other medical domains. The reliability of the MediHall Score was questioned by Reviewers CLg1, 69eN, and Sam2, citing inconsistent results and an overly intuitive hierarchical definition lacking justification. Reviewers h5iV and Sam2 also highlighted the lack of discussion on related work in hallucination detection for healthcare-specific LLMs and insufficient comparisons with existing methods. Additionally, Reviewers 69eN and h5iV criticized the annotation process for being inadequately detailed, raising concerns about reproducibility and ethical considerations such as IRB approval.

No rebuttal was submitted.

---

### Decision · Program_Chairs · 2025-01-22

Reject